# Differentially private Bayesian learning on distributed data

**Mikko Heikkilä**[1]
mikko.a.heikkila@helsinki.fi

**Eemil Lagerspetz**[2]
eemil.lagerspetz@helsinki.fi

**Samuel Kaski**[3]
samuel.kaski@aalto.fi

**Kana Shimizu**[4]
shimizu.kana.g@gmail.com

**Sasu Tarkoma**[2]
sasu.tarkoma@helsinki.fi

**Antti Honkela**[1,5]
antti.honkela@helsinki.fi

[1] Helsinki Institute for Information Technology HIIT,
Department of Mathematics and Statistics, University of Helsinki
[2] Helsinki Institute for Information Technology HIIT,
Department of Computer Science, University of Helsinki
[3] Helsinki Institute for Information Technology HIIT,
Department of Computer Science, Aalto University
[4] Department of Computer Science and Engineering, Waseda University
[5] Department of Public Health, University of Helsinki

## Abstract

Many applications of machine learning, for example in health care, would benefit from methods that can guarantee privacy of data subjects. Differential privacy (DP) has become established as a standard for protecting learning results. The standard DP algorithms require a single trusted party to have access to the entire data, which is a clear weakness, or add prohibitive amounts of noise. We consider DP Bayesian learning in a distributed setting, where each party only holds a single sample or a few samples of the data. We propose a learning strategy based on a secure multi-party sum function for aggregating summaries from data holders and the Gaussian mechanism for DP. Our method builds on an asymptotically optimal and practically efficient DP Bayesian inference with rapidly diminishing extra cost.

## 1 Introduction

Differential privacy (DP) [9, 11] has recently gained popularity as the theoretically best-founded means of protecting the privacy of data subjects in machine learning. It provides rigorous guarantees against breaches of individual privacy that are robust even against attackers with access to additional side information. DP learning methods have been proposed e.g. for maximum likelihood estimation [24], empirical risk minimisation [5] and Bayesian inference [e.g. 8, 13, 16, 17, 19, 25, 29]. There are DP versions of most popular machine learning methods, including linear regression [16, 28], logistic regression [4], support vector machines [5], and deep learning [1].

Almost all existing DP machine learning methods assume that some trusted party has unrestricted access to all the data in order to add the necessary amount of noise needed for the privacy guarantees.

This is a highly restrictive assumption for many applications, e.g. for learning with data on mobile devices, and creates huge privacy risks through a potential single point of failure.

In this paper we introduce a general strategy for DP Bayesian learning in the distributed setting with minimal overhead. Our method builds on the asymptotically optimal sufficient statistic perturbation mechanism [13, 16] and shares its asymptotic optimality. The method is based on a DP secure multi-party communication (SMC) algorithm, called Distributed Compute algorithm (DCA), for achieving DP in the distributed setting. We demonstrate good performance of the method on DP Bayesian inference using linear regression as an example.

## 1.1 Our contribution

We propose a general approach for privacy-sensitive learning in the distributed setting. Our approach combines SMC with DP Bayesian learning methods, originally introduced for the non-distributed setting including a trusted party, to achieve DP Bayesian learning in the distributed setting.

To demonstrate our framework in practice, we combine the Gaussian mechanism for $(\epsilon, \delta)$-DP with efficient DP Bayesian inference using sufficient statistics perturbation (SSP) and an efficient SMC approach for secure distributed computation of the required sums of sufficient statistics. We prove that the Gaussian SSP is an efficient $(\epsilon, \delta)$-DP Bayesian inference method and that the distributed version approaches this quickly as the number of parties increases. We also address the subtle challenge of normalising the data privately in a distributed manner, required for the proof of DP in distributed DP learning.

## 2 Background

### 2.1 Differential privacy

Differential privacy (DP) [11] gives strict, mathematically rigorous guarantees against intrusions on individual privacy. A randomised algorithm is differentially private if its results on adjacent data sets are likely to be similar. Here adjacency means that the data sets differ by a single element, i.e., the two data sets have the same number of samples, but they differ on a single one. In this work we utilise a relaxed version of DP called $(\epsilon, \delta)$-DP [9, Definition 2.4].

**Definition 2.1.** A randomised algorithm $\mathcal{A}$ is $(\epsilon, \delta)$-DP, if for all $\mathcal{S} \subseteq$ Range $(\mathcal{A})$ and all adjacent data sets $D, D'$,

$$P(\mathcal{A}(D) \in \mathcal{S}) \leq \exp(\epsilon) P(\mathcal{A}(D') \in \mathcal{S}) + \delta.$$

The parameters $\epsilon$ and $\delta$ in Definition 2.1 control the privacy guarantee: $\epsilon$ tunes the amount of privacy (smaller $\epsilon$ means stricter privacy), while $\delta$ can be interpreted as the proportion of probability space where the privacy guarantee may break down.

There are several established mechanisms for ensuring DP. We use the Gaussian mechanism [9, Theorem 3.22]. The theorem says that given a numeric query $f$ with $\ell_2$-sensitivity $\Delta_2(f)$, adding noise distributed as $N(0, \sigma^2)$ to each output component guarantees $(\epsilon, \delta)$-DP, when

$$\sigma^2 > 2 \ln(1.25/\delta)(\Delta_2(f)/\epsilon)^2. \tag{1}$$

Here, the $\ell_2$-sensitivity of a function $f$ is defined as

$$\Delta_2(f) = \sup_{D \sim D'} \|f(D) - f(D')\|_2, \tag{2}$$

where the supremum is over all adjacent data sets $D, D'$.

### 2.2 Differentially private Bayesian learning

Bayesian learning provides a natural complement to DP because it inherently can handle uncertainty, including uncertainty introduced to ensure DP [26], and it provides a flexible framework for data modelling.

Three distinct types of mechanisms for DP Bayesian inference have been proposed:

1. Drawing a small number of samples from the posterior or an annealed posterior [7, 25];

2. Sufficient statistics perturbation (SSP) of an exponential family model [13, 16, 19]; and

3. Perturbing the gradients in gradient-based MCMC [25] or optimisation in variational inference [17].

For models where it applies, the SSP approach is asymptotically efficient [13, 16], unlike the posterior sampling mechanisms. The efficiency proof of [16] can be generalised to $(\epsilon, \delta)$-DP and Gaussian SSP as shown in the Supplementary Material.

The SSP (#2) and gradient perturbation (#3) mechanisms are of similar form in that the DP mechanism ultimately computes a perturbed sum

$$z = \sum_{i=1}^{N} z_i + \eta \tag{3}$$

over quantities $z_i$ computed for different samples $i = 1, \ldots, N$, where $\eta$ denotes the noise injected to ensure the DP guarantee. For SSP [13, 16, 19], the $z_i$ are the sufficient statistics of a particular sample, whereas for gradient perturbation [17, 25], the $z_i$ are the clipped per-sample gradient contributions. When a single party holds the entire data set, the sum $z$ in Eq. (3) can be computed easily, but the case of distributed data makes things more difficult.

## 3 Secure and private learning with distributed data

Let us assume there are $N$ data holders (called clients in the following), who each hold a single data sample. We would like to use the aggregate data for learning, but the clients do not want to reveal their data as such to anybody else. The main problem with the distributed setting is that if each client uses a trusted aggregator (TA) DP technique separately, the noise $\eta$ in Eq. (3) is added by each client, increasing the total noise variance by a factor of $N$ compared to the non-distributed single TA setting, effectively reducing to naive input perturbation. To reduce the noise level without compromising on privacy, the individual data samples need to be combined without directly revealing them to anyone.

Our solution to this problem uses an SMC approach based on a form of secret sharing: each client sends their term of the sum, split to separate messages, to $M$ servers such that together the messages sum up to the desired value, but individually they are just random noise. This can be implemented efficiently using a fixed-point representation of real numbers which allows exact cancelling of the noise in the addition. Like any secret sharing approach, this algorithm is secure as long as not all $M$ servers collude. Cryptography is only required to secure the communication between the client and the server. Since this does not need to be homomorphic as in many other protocols, faster symmetric cryptography can be used for the bulk of the data. We call this the Distributed Compute Algorithm (DCA), which we introduce next in detail.

### 3.1 Distributed compute algorithm (DCA)

In order to add the correct amount of noise while avoiding revealing the unperturbed data to any single party, we combine an encryption scheme with the Gaussian mechanism for DP as illustrated in Fig. 1(a). Each individual client adds a small amount of Gaussian noise to his data, resulting in the aggregated noise to be another Gaussian with large enough variance. The details of the noise scaling are presented in the Section 3.1.2.

The scheme relies on several independent aggregators, called Compute nodes (Algorithm 1). At a general level, the clients divide their data and some blinding noise into shares that are each sent to one Compute. After receiving shares from all clients, each Compute decrypts the values, sums them and broadcasts the results. The final results can be obtained by summing up the values from all Computes, which cancels the blinding noise.

### 3.1.1 Threat model

We assume there are at most $T$ clients who may collude to break the privacy, either by revealing the noise they add to their data samples or by abstaining from adding the noise in the first place. The rest are honest-but-curious (HbC), i.e., they will take a peek at other people's data if given the chance, but they will follow the protocol.

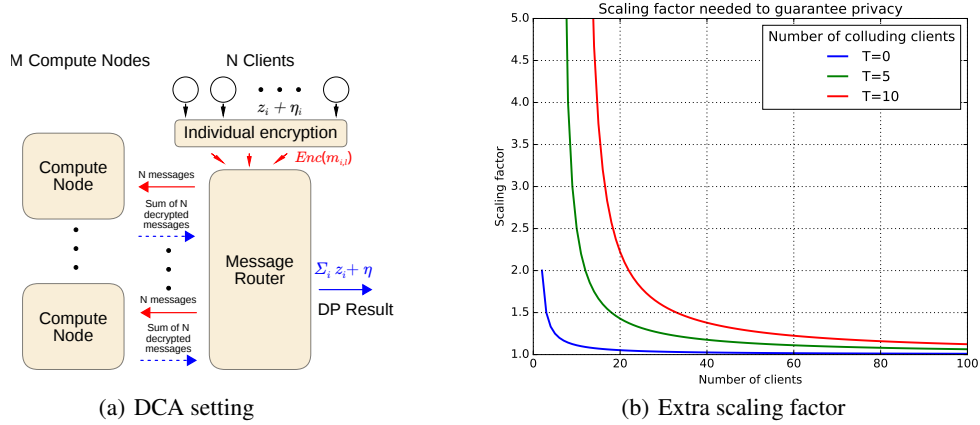

(a) DCA setting           (b) Extra scaling factor

Figure 1: 1(a): Schematic diagram of the Distributed Compute algorithm (DCA). Red refers to encrypted values, blue to unencrypted (but blinded or DP) values. 1(b) Extra scaling factor needed for the noise in the distributed setting with $T$ colluding clients as compared to the trusted aggregator setting.

---

**Algorithm 1** Distributed Compute Algorithm for distributed summation with independent Compute nodes

---

**Input:** $d$-dimensional vectors $\mathbf{z}_i$ held by clients $i \in \{1, \ldots, N\}$;
    Distributed Gaussian mechanism noise variances $\sigma_j^2$, $j = 1, \ldots, d$ (public);
    Number of parties $N$ (public);
    Number of Compute nodes $M$ (public);

**Output:** Differentially private sum $\sum_{i=1}^{N} (\mathbf{z}_i + \boldsymbol{\eta}_i)$, where $\boldsymbol{\eta}_i \sim \mathcal{N}(0, \text{diag}(\sigma_j^2))$

1: Each client $i$ simulates $\boldsymbol{\eta}_i \sim \mathcal{N}(0, \text{diag}(\sigma_j^2))$ and $M - 1$ vectors $\mathbf{r}_{i,k}$ of uniformly random fixed-point data with $\mathbf{r}_{i,M} = -\sum_{k=1}^{M-1} \mathbf{r}_{i,k}$ to ensure that $\sum_{k=1}^{M} \mathbf{r}_{i,k} = \mathbf{0}_d$ (a vector of zeros).
2: Each client $i$ computes the messages $\mathbf{m}_{i,1} = \mathbf{z}_i + \boldsymbol{\eta}_i + \mathbf{r}_{i,1}$, $\mathbf{m}_{i,k} = \mathbf{r}_{i,k}$, $k = 2, \ldots M$, and sends them securely to the corresponding Compute $k$.
3: After receiving messages from all of the clients, Compute $k$ decrypts the values and broadcasts the noisy aggregate sums $\mathbf{q}_k = \sum_{i=1}^{N} \mathbf{m}_{i,k}$. A final aggregator will then add these to obtain $\sum_{k=1}^{M} \mathbf{q}_k = \sum_{i=1}^{N} (\mathbf{z}_i + \boldsymbol{\eta}_i)$.

---

To break the privacy of individual clients, all Compute nodes need to collude. We therefore assume that at least one Compute node follows the protocol. We further assume that all parties have an interest in the results and hence will not attempt to pollute the results with invalid values.

### 3.1.2 Privacy of the mechanism

In order to guarantee that the sum-query results returned by Algorithm 1 are DP, we need to show that the variance of the aggregated Gaussian noise is large enough.

**Theorem 1** (Distributed Gaussian mechanism). *If at most $T$ clients collude or drop out of the protocol, the sum-query result returned by Algorithm 1 is $(\epsilon, \delta)$-DP, when the variance of the added noise $\sigma_j^2$ fulfils*

$$\sigma_j^2 \geq \frac{1}{N - T - 1} \sigma_{j,std}^2,$$

*where $N$ is the number of clients and $\sigma_{j,std}^2$ is the variance of the noise in the standard $(\epsilon, \delta)$-DP Gaussian mechanism given in Eq. (1).*

*Proof.* See Supplement.     □

In the case of all HbC clients, $T = 0$. The extra scaling factor increases the variance of the DP, but this factor quickly approaches 1 as the number of clients increases, as can be seen from Figure 1(b).

### 3.1.3 Fault tolerance

The Compute nodes need to know which clients' contributions they can safely aggregate. This feature is simple to implement e.g. with pairwise-communications between all Compute nodes. In order to avoid having to start from scratch due to insufficient noise for DP, the same strategy used to protect against colluding clients can be utilized: when $T > 0$, at most $T$ clients in total can drop or collude and the scheme will still remain private.

### 3.1.4 Computational scalability

Most of the operations needed in Algorithm 1 are extremely fast: encryption and decryption can use fast symmetric algorithms such as AES (using slower public key cryptography just for the key of the symmetric system) and the rest is just integer additions for the fixed point arithmetic. The likely first bottlenecks in the implementation would be caused by synchronisation when gathering the messages as well as the generation of cryptographically secure random vectors $\mathbf{r}_{i,k}$.

## 3.2 Differentially private Bayesian learning on distributed data

In order to perform DP Bayesian learning securely in the distributed setting, we use DCA (Algorithm 1) to compute the required data summaries that correspond to Eq. (3). In this Section we consider how to combine this scheme with concrete DP learning methods introduced for the trusted aggregator setting, so as to provide a wide range of possibilities for performing DP Bayesian learning securely with distributed data.

The aggregation algorithm is most straightforward to apply to the SSP method [13, 16] for exact and approximate posterior inference on exponential family models. [13] and [16] use Laplacian noise to guarantee $\epsilon$-DP, which is a stricter form of privacy than the $(\epsilon, \delta)$-DP used in DCA [9]. We consider here only $(\epsilon, \delta)$-DP version of the methods, and discuss the possible Laplace noise mechanism further in Section 7. The model training in this case is done in a single iteration, so a single application of Algorithm 1 is enough for learning. We consider a more detailed example in Section 3.2.1.

We can also apply DCA to DP variational inference [17, 19]. These methods rely on possibly clipped gradients or expected sufficient statistics calculated from the data. Typically, each training iteration would use only a mini-batch instead of the full data. To use variational inference in the distributed setting, an arbitrary party keeps track of the current (public) model parameters and the privacy budget, and asks for updates from the clients.

At each iteration, the model trainer selects a random mini-batch of fixed public size from the available clients and sends them the current model parameters. The selected clients then calculate the clipped gradients or expected sufficient statistics using their data, add noise to the values scaled reflecting the batch size, and pass them on using DCA. The model trainer receives the decrypted DP sums from the output and updates the model parameters.

### 3.2.1 Distributed Bayesian linear regression with data projection

As an empirical example, we consider Bayesian linear regression (BLR) with data projection in the distributed setting. The standard BLR model depends on the data only through sufficient statistics and the approach discussed in Section 3.2 can be used in a straightforward manner to fit the model by running a single round of DCA.

The more efficient BLR with projection (Algorithm 2) [16] reduces the data range, and hence sensitivity, by non-linearly projecting all data points inside stricter bounds, which translates into less added noise. We can select the bounds to optimize bias vs. DP noise variance. In the distributed setting, we need to run an additional round of DCA and use some privacy budget to estimate data standard deviations (stds). However, as shown by the test results (Figures 2 and 3), this can still achieve significantly better utility with a given privacy level.

The assumed bounds in Step 1 of Algorithm 2 would typically be available from general knowledge of the data. The initial projection in Step 1 ensures the privacy of the scheme even if the bounds are invalid for some samples. We determine the optimal final projection thresholds $p_j$ in Step 3 using the same general approach as [16]: we create an auxiliary data set of equal size as the original with data

---

**Algorithm 2** Distributed linear regression with projection

---

**Input:** Data and target values $(x_{ij}, y_i), j = 1, \ldots, d$ held by clients $i \in \{1, \ldots, N\}$;
    Number of clients $N$ (public);
    Assumed data and target bounds $(-c_j, c_j), j = 1, \ldots, d+1$ (public);
    Privacy budget $(\epsilon, \delta)$ (public);

**Output:** DP BLR model sufficient statistics of projected data $\sum_{i=1}^{N} \hat{\mathbf{x}}_i \hat{\mathbf{x}}_i^T + \boldsymbol{\eta}^{(1)}, \sum_{i=1}^{N} \hat{\mathbf{x}}_i^T \hat{y}_i + \boldsymbol{\eta}^{(2)},$
    calculated using projection to estimated optimal bounds

 1: Each client projects his data to the assumed bounds $(-c_j, c_j) \; \forall j$.
 2: Calculate marginal std estimates $\sigma^{(1)}, \ldots, \sigma^{(d+1)}$ by running Algorithm 1 using the assumed bounds for sensitivity and a chosen share of the privacy budget.
 3: Estimate optimal projection thresholds $p_j, j = 1, \ldots, d+1$ as fractions of std on auxiliary data. Each client then projects his data to the estimated optimal bounds $(-p_j \sigma^{(j)}, p_j \sigma^{(j)}), j = 1, \ldots, d+1$.
 4: Aggregate the unique terms in the DP sufficient statistics by running Algorithm 1 using the estimated optimal bounds for sensitivity and the remaining privacy budget, and combine the DP result vectors into the symmetric $d \times d$ matrix and $d$-dimensional vector of DP sufficient statistics.

---

generated as

$$x_i \sim N(0, I_d) \tag{4}$$

$$\beta \sim N(0, \lambda_0 I) \tag{5}$$

$$y_i | x_i \sim N(x_i^T \beta, \lambda). \tag{6}$$

We then perform grid search on the auxiliary data with varying thresholds to find the one providing optimal prediction performance. The source code for our implementation is available through GitHub[1] and a more detailed description can be found in the Supplement.

## 4 Experimental setup

We demonstrate the secure DP Bayesian learning scheme in practice by testing the performance of the BLR with data projection, the implementation of which was discussed in Section 3.2.1, along with the DCA (Algorithm 1) in the all HbC clients distributed setting ($T = 0$).

With the DCA our primary interest is scalability. In the case of BLR implementation, we are mostly interested in comparing the distributed algorithm to the trusted aggregator version as well as comparing the performance of the straightforward BLR to the variant using data projection, since it is not clear a priori if the extra cost in privacy necessitated by the projection in the distributed setting is offset by the reduced noise level.

We use simulated data for the DCA scalability testing, and real data for the BLR tests. As real data, we use the Wine Quality [6] (split into white and red wines) and Abalone data sets from the UCI repository[18], as well as the Genomics of Drug Sensitivity in Cancer (GDSC) project data [2]. The measured task in the GDSC data is to predict drug sensitivity of cancer cell lines from gene expression data (see Supplement for a more detailed description). The datasets are assumed to be zero-centred. This assumption is not crucial but is done here for simplicity; non-zero data means can be estimated like the marginal stds at the cost of some added noise (see Section 3.2.1).

For estimating the marginal std, we also need to assume bounds for the data. For unbounded data, we can enforce arbitrary bounds simply by projecting all data inside the chosen bounds, although very poor choice of bounds will lead to poor performance. With real distributed data, the assumed bounds could differ from the actual data range. In the UCI tests we simulate this effect by scaling each data dimension to have a range of length 10, and then assuming bounds of $[-7.5, 7.5]$, i.e., the assumed bounds clearly overestimate the length of the true range, thus adding more noise to the results. The actual scaling chosen here is arbitrary. With the GDSC data, the true ranges are mostly known due to the nature of the data (see Supplement).

|        | N=$10^2$ | N=$10^3$ | N=$10^4$ | N=$10^5$ |
|--------|----------|----------|----------|----------|
| d=10   | 1.72     | 1.89     | 2.99     | 8.58     |
| d=$10^2$ | 2.03   | 2.86     | 12.36    | 65.64    |
| d=$10^3$ | 3.43   | 10.56    | 101.2    | 610.55   |
| d=$10^4$ | 15.30  | 84.95    | 994.96   | 1592.29  |

Table 1: DCA experiment average runtimes in seconds with 5 repeats, using M=10 Compute nodes, N clients and vector length d.

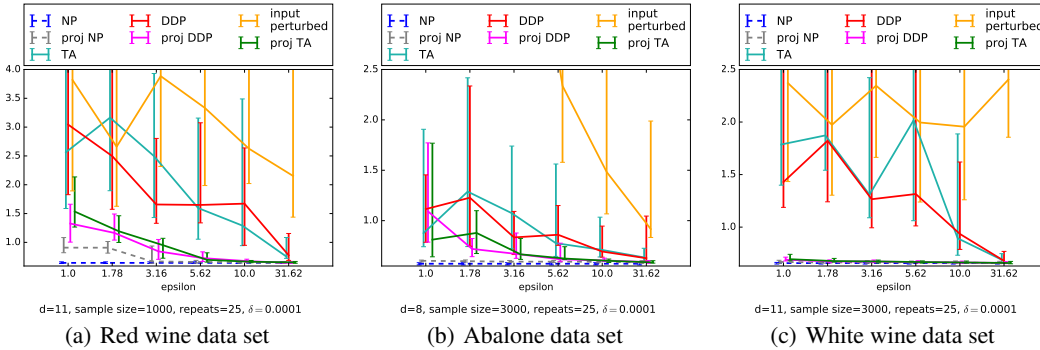

| (a) Red wine data set | (b) Abalone data set | (c) White wine data set |

Figure 2: Median of the predictive accuracy measured on mean absolute error (MAE) on several UCI data sets with error bars denoting the interquartile range (lower is better). The performance of the distributed methods (DDP, DDP proj) is indistinguishable from the corresponding undistributed algorithms (TA, TA proj) and the projection (proj TA, proj DDP) can clearly be beneficial for prediction performance. NP refers to non-private version, TA to the trusted aggregator setting, DDP to the distributed scheme.

The optimal projection thresholds are searched for using 10 (GDSC) or 20 (UCI) repeats on a grid with 20 points between 0.1 and 2.1 times the std of the auxiliary data set. In the search we use one common threshold for all data dimensions and a separate one for the target.

For accuracy measure, we use prediction accuracy on a separate test data set. The size of the test set for UCI in Figure 2 is 500 for red wine, 1000 for white wine, and 1000 for abalone data. The test set size for GDSC in Figure 3 is 100. For UCI, we compare the median performance measured on mean absolute error over 25 cross-validation (CV) runs, while for GDSC we measure mean prediction accuracy to sensitive vs insensitive with Spearman's rank correlation on 25 CV runs. In both cases, we use input perturbation [11] and the trusted aggregator setting as baselines.

## 5   Results

Table 1 shows runtimes of a distributed Spark implementation of the DCA algorithm. The timing excludes encryption, but running AES for the data of the largest example would take less than 20 s on a single thread on a modern CPU. The runtime modestly increases as $N$ or $d$ is increased. This suggests that the prototype is reasonably scalable. Spark overhead sets a lower bound runtime of approximately 1 s for small problems. For large $N$ and $d$, sequential communication at the 10 Compute threads is the main bottleneck. Larger $N$ could be handled by introducing more Compute nodes and clients only communicating with a subset of them.

Comparing the results on predictive error with and without projection (Fig. 2 and Fig. 3), it is clear that despite incurring extra privacy cost for having to estimate the marginal standard deviations, using the projection can improve the results markedly with a given privacy budget.

The results also demonstrate that compared to the trusted aggregator setting, the extra noise added due to the distributed setting with HbC clients is insignificant in practice as the results of the distributed and trusted aggregator algorithms are effectively indistinguishable.

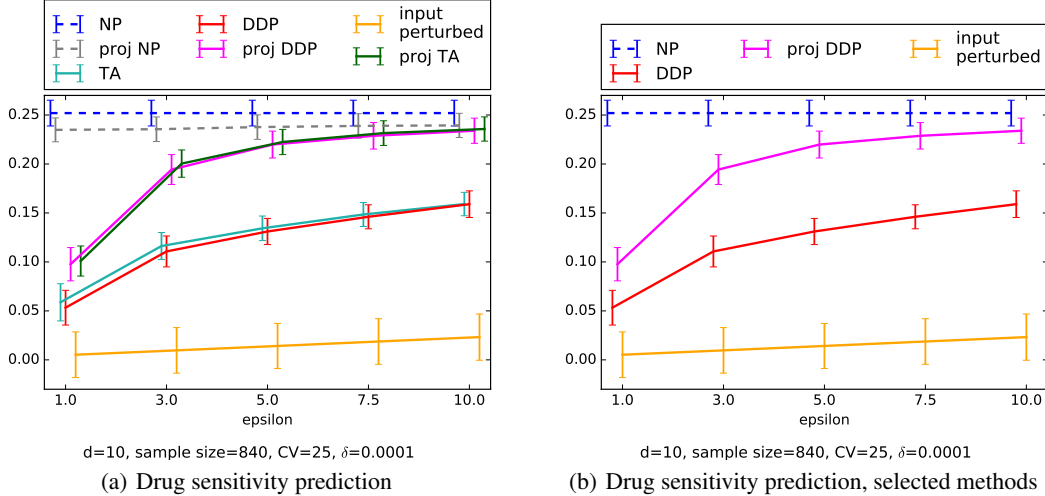

d=10, sample size=840, CV=25, $\delta$=0.0001

(a) Drug sensitivity prediction

d=10, sample size=840, CV=25, $\delta$=0.0001

(b) Drug sensitivity prediction, selected methods

Figure 3: Mean drug sensitivity prediction accuracy on GDSC dataset with error bars denoting standard deviation over CV runs (higher is better). Distributed results (DDP, proj DDP) do not differ markedly from the corresponding trusted aggregator (TA, proj TA) results. The projection (proj TA, proj DDP) is clearly beneficial for performance. The actual sample size varies between drugs. NP refers to non-private version, TA to the trusted aggregator setting, DDP to the distributed scheme.

# 6 Related work

The idea of distributed private computation through addition of noise generated in a distributed manner was first proposed by Dwork et al. [10]. However, to the best of our knowledge, there is no prior work on secure DP Bayesian statistical inference in the distributed setting.

In machine learning, [20] presented the first method for aggregating classifiers in a DP manner, but their approach is sensitive to the number of parties and sizes of the data sets held by each party and cannot be applied in a completely distributed setting. [21] improved upon this by an algorithm for distributed DP stochastic gradient descent that works for any number of parties. The privacy of the algorithm is based on perturbation of gradients which cannot be directly applied to the efficient SSP mechanism. The idea of aggregating classifiers was further refined in [15] through a method that uses an auxiliary public data set to improve the performance.

The first practical method for implementing DP queries in a distributed manner was the distributed Laplace mechanism presented in [22]. The distributed Laplace mechanism could be used instead of the Gaussian mechanism if pure $\epsilon$-DP is required, but the method, like those in [20, 21], needs homomorphic encryption which is computationally more demanding, especially for high-dimensional data.

There is a wealth of literature on secure distributed computation of DP sum queries as reviewed in [14]. The methods of [23, 2, 3, 14] also include different forms of noise scaling to provide collusion resistance and/or fault tolerance, where the latter requires a separate recovery round after data holder failures which is not needed by DCA. [12] discusses low level details of an efficient implementation of the distributed Laplace mechanism.

Finally, [27] presents several proofs related to the SMC setting and introduce a protocol for generating approximately Gaussian noise in a distributed manner. Compared to their protocol, our method of noise addition is considerably simpler and faster, and produces exactly instead of approximately Gaussian noise with negligible increase in noise level.

# 7 Discussion

We have presented a general framework for performing DP Bayesian learning securely in a distributed setting. Our method combines a practical SMC method for calculating secure sum queries with efficient Bayesian DP learning techniques adapted to the distributed setting.

DP methods are based on adding sufficient noise to effectively mask the contribution of any single sample. The extra loss in accuracy due to DP tends to diminish as the number of samples increases and efficient DP estimation methods converge to their non-private counterparts as the number of samples increases [13, 16]. A distributed DP learning method can significantly help in increasing the number of samples because data held by several parties can be combined thus helping make DP learning significantly more effective.

Considering the DP and the SMC components separately, although both are necessary for efficient privacy-aware learning, it is clear that the choice of method to use for each sub-problem can be made largely independently. Assessing these separately, we can therefore easily change the privacy mechanism from the Gaussian used in Algorithm 1 to the Laplace mechanism, e.g. by utilising one of the distributed Laplace noise addition methods presented in [14] to obtain a pure $\epsilon$-DP method. If need be, the secure sum algorithm in our method can also be easily replaced with one that better suits the security requirements at hand.

While the noise introduced for DP will not improve the performance of an otherwise good learning algorithm, a DP solution to a learning problem can yield better results if the DP guarantees allow access to more data than is available without privacy. Our distributed method can further help make this more efficient by securely and privately combining data from multiple parties.

**Acknowledgements**

This work was funded by the Academy of Finland [Centre of Excellence COIN and projects 259440, 278300, 292334, 294238, 297741, 303815, 303816], the Japan Agency for Medical Research and Development (AMED), and JST CREST [JPMJCR1688].

## Footnotes

[1] `https://github.com/DPBayes/dca-nips2017`

[2] `http://www.cancerrxgene.org/`, release 6.1, March 2017

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
