[Supplementary Material]

# Supplement to "Differentially private Bayesian learning on distributed data"

**Mikko Heikkilä, Eemil Lagerspetz, Samuel Kaski,**
**Kana Shimizu, Sasu Tarkoma, and Antti Honkela**

This supplement contains proofs and extra discussion omitted from the main text.

## 1 Privacy and fault tolerance

**Theorem 1** (Distributed Gaussian mechanism). *If at most $T$ clients collude or drop out of the protocol, the sum-query result returned by Algorithm 1 is differentially private, when the variance of the added noise $\sigma_j^2$ fulfils*

$$\sigma_j^2 \geq \frac{1}{N - T - 1}\sigma_{j,std}^2,$$

*where $N$ is the number of clients and $\sigma_{j,std}^2$ is the variance of the noise in the standard Gaussian mechanism given in Eq. (1).*

*Proof.* Using the property that a sum of independent Gaussian variables is another Gaussian with variance equal to the sum of the component variances, we can divide the total noise equally among the $N$ clients.

However, in the distributed setting even with all honest-but-curious clients, there is an extra scaling factor needed compared to the standard DP. Since each client knows the noise values she adds to the data, she can also remove them from the aggregate values. In other words, privacy then has to be guaranteed by the noise the remaining $N - 1$ clients add to the data. If we further assume the possibility of $T$ colluding clients, then the noise from $N - T - 1$ clients must be sufficient to guarantee the privacy.

The added noise can therefore be calculated from the inequality

$$\sum_{i=1}^{N-T-1} \sigma_j^2 \geq \sigma_{j,std}^2 \tag{1}$$

$$\Leftrightarrow \sigma_j^2 \geq \frac{1}{N - T - 1}\sigma_{j,std}^2. \tag{2}$$

$\square$

## 2 Bayesian linear regression

In the following, we denote the $d$-dimensional input data for the $i$th observation by $\mathbf{x}_i$, the scalar target values by $y_i$, and the whole $d + 1-$dimensional dataset by $D_i = (\mathbf{x}_i, y_i)$. We assume all variable-wise expectations to be zeroes for simplicity. For $n$ observations, we denote the sufficient statistics by $n\overline{xx} = \sum_{i=1}^n \mathbf{x}_i\mathbf{x}_i^T$ and $n\overline{xy} = \sum_{i=1}^n \mathbf{x}_i y_i$.

For the regression, we assume that

$$y_i|\mathbf{x}_i \sim N(\mathbf{x}_i^T\beta, \lambda I), i = 1, \dots, n \tag{3}$$
$$\beta \sim N(0, \lambda_0 I), \tag{4}$$

where we want to learn the posterior over $\beta$, and $\lambda, \lambda_0$ are hyperparameters (set to 1 in the tests). The posterior can be solved analytically to give

$$\beta | \mathbf{y}, \mathbf{x} \sim N(\hat{\mu}, \hat{\Lambda}), \tag{5}$$

$$\hat{\Lambda} = \lambda_0 I + \lambda n \overline{xx}, \tag{6}$$

$$\hat{\mu} = \hat{\Lambda}^{-1}(\lambda n \overline{xy}). \tag{7}$$

The predicted mean values from the model are $\hat{y} = \mathbf{x}^T \hat{\mu}$.

The DP sufficient statistics are given by $n\hat{xx} = n\overline{xx} + \eta_{xx}, n\hat{xy} = n\overline{xy} + \eta_{xy}$, where $\eta_{xx}, \eta_{xy}$ consist of suitably scaled Gaussian noise added independently to each dimension. In total, there are $d(d+1)/2 + d$ parameters in the combined sufficient statistic, since $n\overline{xx}$ is a symmetric matrix.

The main idea in the data projection is simply to project the data into some reduced range. Since the noise level is determined by the sensitivity of the data, reducing the sensitivity by limiting the data range translates into less added noise.

With projection threshold $c$, the projection of data $x_i$ is given by

$$\breve{x}_i = \max(-c, \min(x_i, c)). \tag{8}$$

This data projection obviously discards information, but in various problems it can be beneficial to disregard some information in the data in order to achieve less noisy estimates of the model parameters. From the bias-variance trade-off point of view, this can be seen as increasing the bias while reducing the variance. The optimal trade-off then depends on the actual problem.

To run Algorithm 2 (in the main text), we need to assume initial projection bounds $(c_j, d_j)$ for each dimension $j \in \{1, \ldots, d+1\}$ for the data $(\mathbf{x}_i, y_i)_{i=1}^n$. In the paper we assume bounds of the form $(-c_j, c_j)$. To find good final projection bounds, we first find an optimal projection threshold by a grid search on an auxiliary dataset, that is generated from a BLR model similar to the regression model defined above.

This gives us the projection thresholds in terms of std for each dimension. We then estimate the marginal std for each dimension by using Algorithm 1 (in the main text), to fix the actual projection thresholds. For this the data are assumed to lie on some known bounded interval. In practice, the assumed bounds need to be based on prior information. In case the estimates are negative due to noise, they are set to small positive constants (0.5 in all the tests).

The amount of noise each client needs to add to the output depends partly on the sensitivity of the function in question. The query function we are interested in returns a vector of length $d(d+1)/2 + d$ that contains all the unique terms in the sufficient statistics needed for linear regression.

Let $\mathbf{x}, \mathbf{x}'$ be the mismatching, maximally different elements over adjacent datasets s.t. dimensions $1, \ldots, d$ are the independent variables, and $d+1$ is the target. Assume further that each dimension $j = 1, \ldots, d+1$ is bounded by $(-c_j, c_j)$. The squared sensitivity of the query $f$ is then

$$\Delta_2(f)^2 = ||f(\mathbf{x}) - f(\mathbf{x}')||_2^2 \tag{9}$$

$$= ||(x_j x_k - x_j' x_k', x_j x_{d+1} - x_j' x_{d+1}')_{j=1, k=j}^d||_2^2 \tag{10}$$

$$= \sum_{j=1}^d \sum_{k=j}^d (x_j x_k - x_j' x_k')^2 + \sum_{j=1}^d (x_j x_{d+1} - x_j' x_{d+1}')^2 \tag{11}$$

$$\leq \sum_{j=1}^d (c_j^2)^2 + \sum_{j=1}^d \sum_{k>j}^d (2c_j c_k)^2 + \sum_{j=1}^d (2c_j c_{d+1})^2. \tag{12}$$

We assume $c_j = c_x \forall j = 1, \ldots, d$, so (12) can be further simplified to $d(2d-1)c_x^4 + 4d(c_x c_{d+1})^2$.

## 3  Asymptotic efficiency of the Gaussian mechanism

The asymptotic efficiency of the sufficient statistics perturbation using Laplace mechanism has been proven before [2, 3]. We show corresponding results for the Gaussian mechanism. The proofs

generally follow closely those given in [3]. For convenience, we state the relevant definitions, but mostly focus on those proofs that differ in a non-trivial way from the existing ones for the Laplace mechanism. For the full proofs and related discussion, see [3].

## 3.1 Definition of asymptotic efficiency

**Definition 3.1.** A differentially private mechanism $\mathcal{M}$ is *asymptotically consistent with respect to an estimated parameter* $\theta$ if the private estimates $\hat{\theta}_\mathcal{M}$ given a data set $\mathcal{D}$ converge in probability to the corresponding non-private estimates $\hat{\theta}_{NP}$ as the number of samples, $n = |\mathcal{D}|$, grows without bound, i.e., if for any[1] $\alpha > 0$,

$$\lim_{n\to\infty} \Pr\{\|\hat{\theta}_\mathcal{M} - \hat{\theta}_{NP}\| > \alpha\} = 0.$$

**Definition 3.2.** A differentially private mechanism $\mathcal{M}$ is *asymptotically efficiently private with respect to an estimated parameter* $\theta$, if the mechanism is asymptotically consistent and the private estimates $\hat{\theta}_\mathcal{M}$ converge to the corresponding non-private estimates $\hat{\theta}_{NP}$ at the rate $\mathcal{O}(1/n)$, i.e., if for any $\alpha > 0$ there exist constants $C, N$ such that

$$\Pr\{\|\hat{\theta}_\mathcal{M} - \hat{\theta}_{NP}\| > C/n\} < \alpha$$

for all $n \geq N$.

The first part of Theorem 2 follows closely the corresponding result for the Laplace mechanism [3, Theorem 1]. The theorem shows that the optimal rate for estimating the expectation of exponential family distributions is $\mathcal{O}(1/n)$. This justifies the term asymptotically efficiently private introduced by [3], when we show that sufficient statistics perturbation by the Gaussian mechanism achieves this rate.

**Theorem 2.** *The private estimates $\hat{\theta}_\mathcal{M}$ of an exponential family posterior expectation parameter $\theta$, generated by a differentially private mechanism $\mathcal{M}$ that achieves $(\epsilon, \delta)$-differential privacy for any $\epsilon > 0, \delta \in (0, 1)$, cannot converge to the corresponding non-private estimates $\hat{\theta}_{NP}$ at a rate faster than $1/n$. That is, assuming $\mathcal{M}$ is $(\epsilon, \delta)$-differentially private, there exists no function $f(n)$ such that $\limsup nf(n) = 0$ and for all $\alpha > 0$, there exists a constant $N$ such that*

$$\Pr\{\|\hat{\theta}_\mathcal{M} - \hat{\theta}_{NP}\| > f(n)\} < \alpha$$

*for all $n \geq N$.*

*Proof.* The non-private estimate of an expectation parameter of an exponential family is [1]

$$\hat{\theta}_{NP}|x_1, \ldots, x_n = \frac{n_0 x_0 + \sum_{i=1}^n x_i}{n_0 + n}. \tag{13}$$

The difference of the estimates from two neighbouring data sets differing by one element is

$$(\hat{\theta}_{NP}|\mathcal{D}) - (\hat{\theta}_{NP}|\mathcal{D}') = \frac{x - y}{n_0 + n}, \tag{14}$$

where $x$ and $y$ are the corresponding mismatched elements. Let $\Delta = \max(\|x - y\|)$, and let $\mathcal{D}$ and $\mathcal{D}'$ be neighbouring data sets including these maximally different elements.

Let us assume that there exists a function $f(n)$ such that $\limsup nf(n) = 0$ and for all $\alpha > 0$ there exists a constant $N$ such that

$$\Pr\{\|\hat{\theta}_\mathcal{M} - \hat{\theta}_{NP}\| > f(n)\} < \alpha \tag{15}$$

for all $n \geq N$.

Fix $\alpha > 0$ and choose $M \geq \max(N, n_0)$ such that $f(n) \leq \Delta/4n$ for all $n \geq M$. This implies that

$$\|(\hat{\theta}_{NP}|\mathcal{D}) - (\hat{\theta}_{NP}|\mathcal{D}')\| = \frac{\Delta}{n_0 + n} \geq \frac{\Delta}{2n} \geq 2f(n). \tag{16}$$

Let us define the region $C_\mathcal{D} = \{t \mid \|(\hat{\theta}_{NP}|\mathcal{D}) - t\| < f(n)\}$.

Based on our assumptions we have

$$\Pr\left((\hat{\theta}_{\mathcal{M}}|\mathcal{D}) \in C_{\mathcal{D}}\right) > 1 - \alpha. \tag{17}$$

Combining (16) and (15) we have

$$\Pr\left((\hat{\theta}_{\mathcal{M}}|\mathcal{D}') \in C_{\mathcal{D}}\right) < \alpha \tag{18}$$

which implies

$$\Pr\left((\hat{\theta}_{\mathcal{M}}|\mathcal{D}) \in C_{\mathcal{D}}\right) \leq \exp(\epsilon)\Pr\left((\hat{\theta}_{\mathcal{M}}|\mathcal{D}') \in C_{\mathcal{D}}\right) + \delta. \tag{19}$$

Together these imply that

$$1 - \alpha < \exp(\epsilon)\alpha + \delta \tag{20}$$
$$\Leftrightarrow \quad \delta > 1 - (1 + \exp(\epsilon))\alpha. \tag{21}$$

Since for fixed $\epsilon$, $\lim_{\alpha \to 0} 1 - (1 + \exp(\epsilon))\alpha = 1$, $\mathcal{M}$ cannot be $(\epsilon, \delta)$-differentially private with any $\epsilon$ and $\delta < 1$. $\qquad\square$

Before the next theorem, we prove Lemma 1, which is not used in [3].

**Lemma 1.** *Let $x \in \mathbb{R}^d$, $x \sim N(0, \sigma^2 I)$. The tail probability of the $\ell_1$ norm of $x$ obeys*

$$\Pr(\|x\|_1 \geq t) \leq \frac{d\sigma^2}{\left(t - \sqrt{2/\pi}d\sigma\right)^2}\left(1 - \frac{2}{\pi}\right). \tag{22}$$

*Proof.* $\|x\|_1 = \sum_{i=1}^{d} |x_i| = \sum_{i=1}^{d} y_i$, where $x_i \sim N(0, \sigma^2)$ and $y_i$ follows the half-normal distribution with variance $\sigma^2$.

It is known that $\mathrm{E}[y_i] = \sqrt{2/\pi}\sigma$ and $\mathrm{Var}[y_i] = \sigma^2(1 - 2/\pi)$.

Because $y_i$ are independent, $\mathrm{E}[\|x\|_1] = d\,\mathrm{E}[y_i] = \sqrt{2/\pi}d\sigma$ and $\mathrm{Var}[\|x\|_1] = d\,\mathrm{Var}[y_i] = d\sigma^2(1 - 2/\pi)$.

Setting $a = t - \sqrt{2/\pi}d\sigma$ we have

$$\Pr(\|x\|_1 \geq t) = \Pr\left(\|x\|_1 \geq a + \sqrt{2/\pi}d\sigma\right)$$
$$\leq \Pr\left(\left|\|x\|_1 - \sqrt{2/\pi}d\sigma\right| \geq a\right)$$
$$\leq \frac{d\sigma^2}{\left(t - \sqrt{2/\pi}d\sigma\right)^2}\left(1 - \frac{2}{\pi}\right).$$

where the last inequality follows from Chebyshev's inequality. $\qquad\square$

### 3.1.1 Asymptotic efficiency of Gaussian means

Theorem 3, showing one case of asymptotic efficiency of the Gaussian mechanism, corresponds to [3, Theorem 5], although the proof is somewhat different.

**Theorem 3.** *$(\epsilon, \delta)$-differentially private estimate of the mean of a $d$-dimensional Gaussian variable $x$ bounded by $\|x_i\|_1 \leq B$ in which the Gaussian mechanism is used to perturb the sufficient statistics, is asymptotically efficiently private.*

*Proof.* Following [3, Theorem 3], it is trivial to show that

$$\|\mu_{DP} - \mu_{NP}\|_1 \leq \frac{c}{n}\|\delta\|_1,$$

where $\delta = (\delta_1, \ldots, \delta_d)^T \in \mathbb{R}^D$ with $\delta_j \sim N\left(0, \sigma_j^2\right)$ holds when we utilize the Gaussian mechanism instead of the Laplace mechanism. This allows us to bound the corresponding tail probabilities by using Lemma 1.

Therefore, given $\alpha > 0$, we can guarantee that

$$\Pr\left\{\|\mu_{DP} - \mu_{NP}\|_1 > \frac{C}{n}\right\} \leq \Pr\left\{\frac{1}{n}\|\delta\|_1 > \frac{C}{n}\right\} = \Pr\{\|\delta\|_1 > C\} < \alpha, \qquad (23)$$

by choosing $C$ according to Lemma 1. $\qquad\square$

## 3.2 Asymptotic efficiency of DP linear regression

Theorem 4 that establishes asymptotic efficiency for DP linear regression using the Gaussian mechanism, for the most part follows [3, Theorem 8]. We concentrate here more closely only on the differing parts.

**Theorem 4.** $(\epsilon, \delta)$-*differentially private inference of the posterior mean of the weights of linear regression with the Gaussian mechanism used to perturb the sufficient statistics is asymptotically efficiently private.*

*Proof.* Following the proof of [3, Theorem 7] with minimal changes we have

$$\|\mu_{DP} - \mu_{NP}\|_1 \leq \left\|(\Lambda_0 + \Lambda(n\overline{xx} + \Delta))^{-1}\Lambda\delta\right\|_1$$
$$+ \left\|\left[\left(\frac{1}{n}\Lambda_0 + \Lambda\left(\overline{xx} + \frac{1}{n}\Delta\right)\right)^{-1} - \left(\frac{1}{n}\Lambda_0 + \Lambda\overline{xx}\right)^{-1}\right]\left(\Lambda\overline{xy} + \frac{1}{n}\Lambda_0\beta_0\right)\right\|_1, \quad (24)$$

where $\Delta$ is the noise contribution from the Gaussian mechanism added to the sufficient statistics $\overline{xx}$ (see Section 2 in this supplement).

As in [3, Theorem 7], the first term can be bounded as

$$\left\|(\Lambda_0 + \Lambda(n\overline{xx} + \Delta))^{-1}\Lambda\delta\right\|_1 \leq \frac{c_1}{n}\left\|(\overline{xx})^{-1}\right\|_1 \|\delta\|_1 \qquad (25)$$

where $c_1 > 1$, for large enough $n$.

As done in the proof of Theorem 3, given $\alpha > 0$, Lemma 1 can be used to ensure that

$$\Pr\left\{\|(\Lambda_0 + \Lambda(n\overline{xx} + \Delta))^{-1}\Lambda\delta\|_1 > \frac{C_1}{n}\right\} < \frac{\alpha}{2}, \qquad (26)$$

by choosing a suitable $C_1$.

Again, following [3, Theorem 7], the second term can be bounded as

$$\left\|\left[\left(\frac{1}{n}\Lambda_0 + \Lambda\left(\overline{xx} + \frac{1}{n}\Delta\right)\right)^{-1} - \left(\frac{1}{n}\Lambda_0 + \Lambda\overline{xx}\right)^{-1}\right]\left(\Lambda\overline{xy} + \frac{1}{n}\Lambda_0\beta_0\right)\right\|_1$$
$$\leq \frac{c_2}{n}\left\|(\overline{xx})^{-1}\right\|_1 \|\Delta\|_1\left\|(\overline{xx})^{-1}\right\|_1 \|\overline{xy}\|_1,$$

where, as in Eq. (25), the bound is valid for $c_2 > 1$ as $n$ gets large enough.

$\|\Delta\|_1$ here is the $\ell_1$-norm of the symmetric matrix $\Delta$, that is comprised of a vector of $d(d+1)/2$ unique noise terms, each generated independently from a Normal distribution according to the Gaussian mechanism. Denoting this vector by $\mathbf{v}$, a bound to the matrix norm is given by $\|\Delta\|_1 \leq \|\mathbf{v}\|_1$.

Therefore, given $\alpha > 0$, we can again use Lemma 1 to find a suitable $C_2$ s.t.

$$\Pr\left\{\|\Delta\|_1 > \frac{C_2}{c_2\left\|(\overline{xx})^{-1}\right\|_1^2 \|\overline{xy}\|_1}\right\} \leq \Pr\left\{\|\mathbf{v}\|_1 > \frac{C_2}{c_2\left\|(\overline{xx})^{-1}\right\|_1^2 \|\overline{xy}\|_1}\right\} < \frac{\alpha}{2}. \qquad (27)$$

By combining Eqs. (26) and (27) we get

$$\Pr\left\{\|\mu_{DP} - \mu_{NP}\|_1 > \frac{C_1 + C_2}{n}\right\} < \alpha. \qquad (28)$$

$\qquad\square$

## 4   GDSC dataset description

The data were downloaded from the Genomics of Drug Sensitivity in Cancer (GDSC) project, release 6.1, March 2017, http://www.cancerrxgene.org/. We use gene expression and drug sensitivity data. The gene expression dimensionality is reduced to 10 genes used for the actual prediction task, based on prior information about their mutation counts in cancer (we use the same procedure as [3]). The dataset used for learning contains 940 cell lines and drug sensitivity data for 265 drugs. Some of the values are missing, so the actual number of observations varies between the drugs. We use a test set of size 100 and the rest of the available data for learning.

Since we want to focus on the relative expression of the genes, each data point is normalized to have $\ell_2$-norm of 1. In the distributed setting this can be done by each client without breaching privacy. After the scaling, we know that all dimensions are bounded by $[-1, 1]$, except the target. For the target dimension, the true range varies between the drugs. The average width of the ranges is $8.6$.

We assume a range of [-7.5,7.5] for the marginal std estimation needed for the projection, and use a symmetric bound given by $[-\lceil \max |y| \rceil, \lceil \max |y| \rceil]$ for the non-projected baseline methods (DDP, TA). The exact bound for the baseline methods varies between the drugs while the average is $6.8$. In other words, the projected methods add somewhat more extra noise to the results on average. We also tested the performance using a fixed bound for the non-projected methods as with the UCI data, but the results did not change markedly (not included in the paper).

## Footnotes

[1]We use $\alpha$ in limit expressions instead of usual $\epsilon$ to avoid confusion with $\epsilon$-differential privacy.