[Reviews · NeurIPS 2017]

Reviewer 1



Summary: This paper combines secure multiparty computation ideas with differential privacy (DP) as applied to bayesian learning, in a distributes setting. To expand a bit: Bayesian Learning with DP has been studied recently and there are a few orthogonal approaches to that problem. Some of these use as a privacy primitive the Gaussian noise mechanism. In the centralized setting, or in the presence of a trusted party, these approaches can be used. This work studies a distributed setting where one does not have a single trusted party. The authors propose a mechanism that they call DCA (Distributed compute algorithm) that allows one to compute a noisy sum securely, under their threat model which assumes that while there is no single trusted party, there is a set of compute nodes of which at least one is trusted. Given this algorithm, one can do other things that use noisy sum as a primitive. This includes some approaches to bayesian learning, such as Sufficient Statistics Perturbation and gradient based MCMC methods. Opinion: The summing algorithm is simple and closely related to previous work on the subject such as 23. While the algorithm is definitely different here, as the threat model is different, I think the novelty in this paper is not sufficient to justify acceptance to this conference. I have read the rebuttal. I agree that simplicity is a good thing and do not hold that against the paper. This is an interesting contribution that should be published somewhere, but am unconvinced that it is strong enough for this conference.

Reviewer 2



Title: Differentially private Bayesian learning on distributed data Comments: - This paper develops a method for differential privacy (DP) Bayesian learning in a distributed setting, where data is split up over multiple clients. This differs from the traditional DP Bayesian learning setting, in which a single party has access to the full dataset. The main issue here is that performing DP methods separately on each client would yield too much noise; the goal is then to find a way to add an appropriate amount of noise, without compromising privacy, in this setting. To solve this, the authors introduce a method that combines existing DP Bayesian learning methods with a secure multi-party communication method called the DCA algorithm. Theoretically, this paper shows that the method satisfies differential privacy. It also shows specifically how to combine the method with existing DP methods that run on each client machine. Experiments are shown on Bayesian linear regression, on both synthetic and real data. - Overall, I feel that this method provides a nice contribution to work on Bayesian DP, and extends where one can use these methods to data-distributed settings, which seems like a practical improvement. - I feel that this paper is written quite nicely. It is therefore easy to understand previous work, the new method, and how the new method fits in context with previous work. - One small note: I might’ve missed it, but please make sure the acronym TA (for trusted aggregator, I believe) is defined the paper. I believe “TA” might be used in a few places without definition.

Reviewer 3



1. The problem formulation is perhaps not that practical. Even if it is, the experiments are not designed accordingly to establish the relevance of the formulation for solving some real-world problems. 2. The analysis techniques used herein are also pretty standard in the literature of differential privacy. So I'm not sure how the audience of NIPS would accept such formulation. 3. There is no other criticism to offer for this paper. In fact, the paper could get accepted in some other conference as it is, but perhaps does not meet the high standard that NIPS demands.